# NEK2 Phosphorylates RhoGDI1 to Promote Cell Proliferation, Migration and Invasion Through the Activation of RhoA and Rac1 in Colon Cancer Cells

**DOI:** 10.3390/cells13242072

**Published:** 2024-12-16

**Authors:** Jeewon Lim, Yo-Sep Hwang, Jong-Tae Kim, Hyang-Ran Yoon, Hyo-Min Park, Jahyeong Han, Taeho Kwon, Kyung-Ho Lee, Hee-Jun Cho, Hee-Gu Lee

**Affiliations:** 1Immunotherapy Research Center, Korea Research Institute of Bioscience and Biotechnology, Daejeon 34141, Republic of Korea; ljw8796@kribb.re.kr (J.L.); hys8520@kribb.re.kr (Y.-S.H.); kjtdna@kribb.re.kr (J.-T.K.); yhr1205@kribb.re.kr (H.-R.Y.); wkd910222@kribb.re.kr (H.-M.P.); hanjh@kribb.re.kr (J.H.); 2Department of Biomolecular Science, University of Science and Technology (UST), Daejeon 34113, Republic of Korea; 3Primate Resources Center, Korea Research Institute of Bioscience and Biotechnology, Jeongeup 56216, Republic of Korea; kwon@kribb.re.kr; 4Chemical Biology Research Center, Korea Research Institute of Bioscience and Biotechnology, Cheongju 28644, Republic of Korea; leekh@kribb.re.kr

**Keywords:** NEK2, RhoGDI1, Rho GTPase, proliferation, migration, cancer

## Abstract

Rho guanine nucleotide dissociation inhibitor 1 (RhoGDI1) plays a critical role in regulating the activity of Rho guanosine triphosphatases (GTPases). Phosphorylation of RhoGDI1 dynamically modulates the activation of Rho GTPases, influencing cell proliferation and migration. This study explored the involvement of Never In Mitosis A (NIMA)-related serine/threonine protein kinase 2 (NEK2) in phosphorylating RhoGDI1 and its implications in cancer cell behavior associated with tumor progression. We employed GST pull-down assays and immunoprecipitation to investigate the interaction between NEK2 and RhoGDI1. Truncation fragments identified the region of RhoGDI1 responsible for binding with NEK2. Phosphorylation assays determined the site of NEK2-mediated phosphorylation on RhoGDI1. Functional assays were conducted using overexpression of the RhoGDI1 substitution mutant to assess their impact on cancer cell behavior. NEK2 directly bound to RhoGDI1 and phosphorylated it at Ser174. This phosphorylation event facilitated cancer cell proliferation and motility by activating RhoA and Rac1. The RhoGDI1 aa 112–134 region was critical for the binding to NEK2. Disruption of the NEK2–RhoGDI1 interaction through overexpression of a RhoGDI1 truncated fragment (aa 112–134) led to diminished RhoGDI1 phosphorylation and RhoA/Rac1 activation induced by NEK2, resulting in reduced cancer cell proliferation and migration. Moreover, in vivo studies showed reduced tumor growth and lung metastasis when the NEK2–RhoGDI1 interaction was disrupted. This study indicates that NEK2 promotes the metastatic behaviors of cancer cells by activating RhoA and Rac1 by phosphorylating RhoGDI1.

## 1. Introduction

Rho GTPases are recognized to function in many types of cellular processes, including cell polarity, cytoskeletal rearrangements, gene transcription, proliferation, cell motility, and vesicle trafficking [1]. Dysregulation of Rho GTPase signaling causes malignant transformation, immunological diseases, and neurological abnormalities [2].

Typically, Rho GTPases have a cycle of an active conformation bound to GTP in the membrane and an inactive conformation bound to GDP in the cytosol, in which three regulators are involved [3]. Guanine nucleotide exchange factors (GEFs) activate Rho GTPases by facilitating the exchange of GDP for GTP. Meanwhile, GTPase-activating proteins (GAPs) promote intrinsic hydrolysis of GTP, thereby inactivating Rho GTPase [4,5]. Guanine nucleotide dissociation inhibitors (GDIs) bind to the GDP-bound form of Rho GTPases, sequestering them in the cytosol and preventing them from being located in the membrane and activated by GEFs [6]. Therefore, Rho GTPases are spatiotemporally and dynamically regulated through GDIs, GEFs, and GAPs to determine specific cell behavior [7].

RhoGDI1 is ubiquitously expressed in diverse cells. Most Rho GTPases are in an inactive state by binding to RhoGDIs in the cytosol, but when they are dissociated from RhoGDIs, they can transfer to the membrane and be activated by RhoGEFs. RhoGDIs re-associate with Rho GTPases on the membrane and sequester them back in the cytosol for recycling [8,9]. Therefore, it is important that RhoGDIs associate and dissociate with Rho GTPases, which localize them to the cytosol and the plasma membrane, respectively, in regulating the activity of Rho GTPases. Their association can be regulated by protein–protein interaction and post-translational modification (PTM) such as phosphorylation and dephosphorylation, leading to malignant transformation [6,7]. For instance, 14-3-3 binds directly to RhoGDI1 phosphorylated at Ser174, releasing RhoA, Rac1, and Cdc42, promoting Rho GTPases activation and resulting in increased cancer cell invasion and metastasis [10]. EphB2 receptors interact with EphrinB1 to regulate cell-to-cell adhesion, in which EphrinB1 causes dissociation and activation of RhoA through direct binding to RhoGDI1, thereby promoting cancer cell migration and invasion [11]. Furthermore, the CD146 and ERM complex associates with RhoGDI1, spatially sequestering it from RhoA, and then enhancing melanoma cell migration by activated RhoA [12]. The function of RhoGDIs is also regulated by phosphatase. Protein phosphatase 1B (PPM1B) dephosphorylates RhoGDI1 at Ser174, reduces RhoGDI1 interaction with 14-3-3, thereby inhibiting signaling of RhoA, Rac1, and Cdc42 in breast cancer cell migration [13]. These reports suggest that dysregulation of RhoGDI1-Rho GTPase affinity is associated with malignancy.

Never In Mitosis A (NIMA)-related serine/threonine protein kinase 2 (NEK2) is well known to regulate cell cycle, such as centrosome duplication and separation [14,15], microtubule stabilization [16], kinetochore attachment [17], and spindle assembly checkpoint [18,19]. Thus, dysfunction of NEK2 leads to chromosome instability, tumorigenesis, and cancer progression [20,21,22]. Furthermore, the role of NEK2 has been reported in the migration and invasion of diverse cancer cell lines, including breast, colorectal, liver, lung, and prostate cancer cell [23,24,25,26,27]. Despite the fact that many articles have proposed NEK2 as a therapeutic molecular target in the progression of various cancers [28,29,30], the mechanism underlying NEK2 effect on cell motility is unclear. In this study, we investigated the involvement of NEK2 in phosphorylating RhoGDI1 and its implications in cancer cell behavior associated with tumor malignant progression.

## 2. Materials and Methods

### 2.1. Cell Lines and Reagents

HeLa, Ls174T, LoVo, DLD-1, Caco-2, COLO205, HCT-15, HCT116, HT-29, KM12C, KM12SM, SW480, and SW620 cell lines were obtained from the American Type Culture Collection (ATCC, Manassas, VA, USA). All cells were grown in RPMI (Welgene, Gyeongbuk, Republic of Korea, LM011-05) or DMEM (Welgene, LM001-05) supplemented with 1% antibiotics (Gibco, Waltham, MA, USA, 11570486) and 10% fetal bovine serum (FBS). All cell lines were maintained at 37 °C with 5% CO_2_ in humidified conditions. NCL 00017509, a NEK2 inhibitor, was obtained from Tocris (Bristol, UK, 5150).

### 2.2. Plasmids

Human RhoGDI1 and NEK2 cDNA were ordered from Origene (Rockville, MD, USA, RG200902) and ABM (Richmond, BC, USA, ORF007021). Flag-RhoGDI1 and HA-NEK2A were amplified using PCR and subjected to clone into pCDNA3.1(+) or pLVX-EF1α-IRES-Puro Vector. After substitution mutants and truncation fragments of RhoGDI1 were synthesized from Bioneer (Daejeon, Republic of Korea), the RhoGDI1 mutants and fragments were cloned into vectors, including pGEX-4T-1, pCDNA3.1(+), pEGFP-N2, or pCDH-CMV-MCS-EF1-hygro-mCherry.

### 2.3. Cell Transfection and Lentivirus Infection

Cells were transiently transfected using Lipofectamine 2000 (Invitrogen, Carlsbad, CA, USA) reagent for 24–36 h with indicated plasmid constructs, according to the manufacturer’s instruction. DLD-1 cells were infected with lentiviruses delivering the HA-NEK2 or mock constructs and selected with 4 μg/mL puromycin for 6 passages. HA-NEK2 or mock cell line was infected with lentiviruses delivering the mCherry aa 112–134 or mCherry constructs followed by 200 μg/mL hygromycin selection for 6 passages. HCT116 cells were infected with lentiviruses delivering the shRNAs of NEK2 and selected with 4 μg/mL puromycin for 6 passages.

### 2.4. Pull-Down Assay

For GST pull-down assay, The GSH-Beads (GE Healthcare, Chicago, IL, USA, 17-0756-01) were reacted with recombinant RhoGDI1 (Cytoskeleton, Denver, CO, GDI01) and RhoGDI2 (Novusbio, E Easter Ave, Centennial, CO, USA, NBP1-99054) proteins overnight at 4 °C and then mixed with active His-NEK2 (Thermo Fisher Scientific, Waltham, MA, USA, PV3360) overnight at 4 °C. For His pull-down assay, Ni-NTA agarose beads (QIAGEN, 30210) were reacted with His-NEK2 (Thermo Fisher Scientific, PV3360) overnight at 4 °C and then mixed with GST-tagged wild-type (wt) or truncated fragments of RhoGDI1 purified from *E. coli* lysates overnight at 4 °C. After three washes of the beads, the samples were analyzed by WB.

### 2.5. Immunoprecipitation

Cells were harvested in ice-cold RIPA buffer (50 mM β-glycerophosphate, 50 mM NaCl, 0.1 mM Na_3_VO_4_, 50 mM NaF, 0.5% sodium deoxycholate, 5 mM EDTA, 0.5% NP-40, and 100 mM Tris-HCl pH 7.6) containing protease inhibitor. After quantification of cell lysates, the samples were incubated with anti-RhoGDI1 (Invitrogen, Carlsbad, CA, USA, 51-1000Z) at 4 °C for 24 h and then agarose beads (Santa Cruz Biotechnology, Dallas, TX, USA, sc-2003) at 4 °C for 2 h.

### 2.6. In Vitro Kinase Assay

His-tagged RhoGDI1 were purified from *E.coli* lysates. Ni-NTA beads (QIAGEN, Hilden, Germany, 30210) were reacted with recombinant RhoGDI1 protein, washed three times, and incubated with active NEK2 (Thermo Fisher Scientific, PV3360) and kinase buffer (9802, Cell Signaling Technology, Danvers, MA, USA) containing 200 μM ATP for 1 h at 30 °C. After termination of the reactions with 50 μL sample loading buffer, the mixtures were analyzed with WB.

### 2.7. Rho GTPases Activity Assay

Cells were harvested with Mg2+ lysis/wash buffer (MLB). Activation levels of Rho GTPases were evaluated using Rho assay reagent (14-383) or Rac/Cdc42 assay reagent (14-325) of Millipore (Burlington, MA, USA), according to the manufacturer’s instruction and analyzed by WB with antibodies to Rho GTPases.

### 2.8. Western Blot (WB) Analysis and Antibodies

Samples were separated by 10–15% SDS-PAGE, transferred to PVDF membranes and blocked with 5% skim milk, which were probed with antibodies, including RhoGDI1 (sc-373724), GST (sc-138), RhoA (sc-418), GFP (sc-9996), and β-Actin (sc-47778) of Santa Cruz Biotechnology, RhoGDI1 (S174) (ab74142) and 14-3-3 tau (ab10439) of Abcam (Cambridge, UK), NEK2 (610594), Cdc42 (610929), and Rac1 (610651) of BD Bioscience (Franklin Lakes, NJ, USA), Phospho-(Ser/Thr) (Cell Signaling Technology, 9631), HA-HRP (Roche, Basel, Switzerland, 12013819001), and Flag-HRP (Sigma, St. Louis, MO, USA, A8592).

### 2.9. Wound-Healing Assay

Cells were seeded in Culture-Inserts (Ibidi, Grafelfing, Germany, 80242) on a 6-well plate at 5 × 10^4^/50 μL and incubated for cell attachment. After removing culture-inserts, each well was filled with complete culture medium. At 0, 24, and 48 h following media replacement, cell migration images were captured with microscopy at a 10× objective. ImageJ (ver. 1.54) was used to measure the wound area. Cell migration was calculated by the rate of reduced area compared to the area at T0.

### 2.10. Invasion Assays

Cell invasion was assessed with transwell membrane inserts (Corning, Corning, NY, USA, 3422) and Matrigel (Corning, 354234), as previously described [13].

### 2.11. WST-8 Assay

Cell proliferation was measured using WST-8 reagent (Medifab, Seoul, Republic of Korea, B1007). Briefly, cells were seeded at 5 × 10^3^ cells/well of a 96-well plate, incubated for 48 h, and then treated with 10% WST-8 reagent for 2 h. Absorbance of the wells was measured at 450 nm.

### 2.12. Animal Experiments

The effect on tumor growth and metastasis was performed as previously described [31]. Briefly, Female BALB/c nude mice (6–8 week old) were purchased from Dae Han Bio Link Co., Ltd. (Chungbuk, Republic of Korea) and housed in a specific pathogen-free (SPF) room. For the in vivo tumor growth assay, the mice were randomly divided into three groups (*n* = 8 per group) and were subcutaneously inoculated with 5 × 10^6^ DLD-1 cells expressing the mCherry only or mCherry-RhoGDI1 aa 112–134 with or without NEK2. Then, 4 weeks after implantation, the mice were euthanized and their primary tumors were excised to measure the tumor weight and volume. Tumor volumes (mm^3^) were estimated by the following formula: 0.52 × (length) × (width) × (height). Additionally, tumors were subjected to immunohistochemistry (IHC) staining for Ki-67 and CD31.

In the in vivo tumor metastasis assay, the mice were randomly divided into three groups (*n* = 8 per group) and were intravenously inoculated with 2 × 10^6^ DLD-1 cells expressing the mCherry only or mCherry-RhoGDI1 aa 112–-134 with or without NEK2. Then, 8 weeks after implantation, the mice were euthanized and their lung tissues were excised to count the nodules of the lungs. Additionally, lung tissues were fixed by 4% Paraformaldehyde and stained by hematoxylin and eosin (H&E) staining. All animal experimental instructions were performed in compliance with the approval of the KRIBB Institutional Animal Care and Use Committee (KRIBB-AEC-24004).

### 2.13. Statistics

Student’s *t*-test was used for comparisons of statistical analyses. Quantitative data were shown as means ± standard deviations and considered statistically significant with *p*-value < 0.05.

## 3. Results

### 3.1. NEK2 Interacts with RhoGDI1

NEK2 was previously identified as a RhoGDI1-interacting candidate protein with a proteomic approach [13]. To confirm the NEK2 binding to RhoGDI1, HA-NEK2 was co-transfected with Flag-RhoGDI1 or Flag-RhoGDI2 in HeLa cells, and their interaction was analyzed via co-immunoprecipitation using an HA antibody. The results show specific binding between HA-NEK2 and Flag-RhoGDI1, but not Flag-RhoGDI2 (Figure 1A). Additionally, GST pull-down assays with recombinant His-tagged NEK2 and GST-tagged RhoGDI1 or RhoGDI2 confirmed a direct interaction between NEK2 and RhoGDI1 (Figure 1B). These findings indicate that NEK2 directly binds to RhoGDI1, but not RhoGDI2.

Because NEK2 is known to be overexpressed in human colorectal cancer [28], we first evaluated the expression of NEK2 and RhoGDI1 in various human colon cancer cell lines. Western analysis indicated that NEK2 is abundantly expressed in most colon cancer cell lines, while it is weakly expressed in Ls174T, LoVo, and DLD-1 cells (Figure 1C). Next, we examined that endogenous NEK2 and RhoGDI1 bind to each other by immunoprecipitation with a RhoGDI1 antibody in HT-29 and HCT116 cell lysates. As expected, endogenous NEK2 was detected in RhoGDI1 immune complexes from HT-29 and HCT116 cell lines (Figure 1D), indicating that NEK2 interacts with RhoGDI1 in cells.

To further explore the region of RhoGDI1 involved in its interaction with NEK2, we expressed and purified GST-tagged RhoGDI1 wild type (WT) and three RhoGDI1 truncated fragments using bacterial systems (Figure 2A). His pull-down assays demonstrated that His-NEK2 interacts specifically with GST-RhoGDI1 WT and GST-tagged regions spanning aa 1–134 and aa 68–201 of RhoGDI1, but not with aa 1–67 of RhoGDI1, suggesting the importance of RhoGDI1 aa 68–134 for binding to NEK2 (Figure 2B). To further investigate the region of RhoGDI1 responsible for its binding to NEK2, we generated additional fragments spanning RhoGDI1 aa 68–134 (Figure 2A). Among these fragments, RhoGDI1 aa 112–134 showed the highest interaction with NEK2, while RhoGDI1 aa 68–89 slightly retained its affinity for NEK2, and aa 90–111 did not show any affinity at all (Figure 2C). Moreover, transfection of GFP-tagged RhoGDI1 aa 112–134 into HCT116 cells significantly disrupted the RhoGDI1–NEK2 interaction (Figure 2D). These results highlight the requirement of aa 112–134 of RhoGDI1 for its association with NEK2.

### 3.2. NEK2 Phosphorylates RhoGDI1 at Ser174

Because NEK2 is known to be a Ser/Thr kinase [32], we investigated whether NEK2 phosphorylates RhoGDI1. In vitro kinase assays were conducted using bacterially purified His-RhoGDI1 wild type (WT) with active NEK2, demonstrating that NEK2 could phosphorylate His-RhoGDI1 (Figure 3A). RhoGDI1 has been known to be phosphorylated at Ser34, Ser96, Ser101, Ser174 and Thr7/91 by several kinases [33,34,35,36,37]. Therefore, to identify the phosphorylated residue on RhoGDI1 by NEK2, active NEK2 was reacted with bacterially expressed His-RhoGDI1 WT, S34A, S96A, S101A, S174A and T7/91A mutants. The result shows that NEK2-mediated phosphorylation of RhoGDI1 was entirely abrogated by S174A mutation (Figure 3B). These findings indicate that NEK2 phosphorylates RhoGDI1 at Ser174 in vitro. To elucidate whether NEK2 phosphorylates RhoGDI1 in cells, we transfected with the control or NEK2-expressing vector into the DLD-1 cell line which expresses low levels of endogenous NEK2 compared to the expression levels of NEK2 in HCT116. Subsequent analysis of cell lysates using an anti-RhoGDI1 (phospho S174) antibody revealed increased phosphorylation of RhoGDI1 at Ser174 in NEK2-expressing DLD-1 cells. Conversely, this phosphorylation was reduced upon pharmacological inhibition of NEK2 (Figure 3C). Moreover, both pharmaceutical inhibition and shRNA-mediated knockdown of NEK2 led to decreased levels of phosphorylated RhoGDI1 in HCT116 cells, which express endogenous NEK2 (Figure 3D,E). To explore whether NEK2-mediated phosphorylation of RhoGDI1 was correlated with its interaction with RhoGDI1, we employed a fragment of GFP-RhoGDI1 aa 112–134 designed to bind to NEK2 and disrupt its interaction with endogenous RhoGDI1 (Figure 2D). The GFP vector or GFP-RhoGDI1 aa 112–134 were transiently transfected to NEK2-expressing DLD-1 cells. WB analysis revealed that RhoGDI1 aa 112–134 reduced the phosphorylation of RhoGDI1 by NEK2 overexpression (Figure 3F). Because 14-3-3 directly interacts with RhoGDI1 phosphorylated at Ser174 [10], we verified whether NEK2-mediated phosphorylation of RhoGDI1 promotes the interaction of 14-3-3 with RhoGDI1. Lysates of the control vector or NEK2-expressing DLD-1 cells were subjected to immunoprecipitation with a RhoGDI1 antibody. WB analysis showed that NEK2 overexpression enhanced the interaction with RhoGDI1 and 14-3-3 compared to mock vector expression (Figure 3G). Collectively, these results indicate that NEK2 phosphorylates Ser174 residue of RhoGDI1 in cells.

### 3.3. NEK2 Promotes RhoA and Rac1 Activation by Interacting with RhoGDI1

The phosphorylation of RhoGDI1 at Ser174 facilitates its dissociation from Rho GTPases and regulates their activity [10,13,35]. Because NEK2 promoted the phosphorylation of RhoGDI1 at Ser174, we investigated the impact of NEK2 on the activity of Rho GTPases. The lysates of DLD1 cells expressing control vector or NEK2 were subjected to pull-down assays using RBD or PBD agarose beads. In the case of NEK2-expressing DLD1 cells, there was a notable increase in the levels of GTP-bound active RhoA and Rac1 compared to the control cells. Interestingly, the level of GTP-bound active Cdc42 was not affected by NEK2 overexpression (Figure 4A). The treatment of the NEK2 inhibitor effectively counteracted the heightened activities of RhoA and Rac1 induced by NEK2 overexpression (Figure 4B). Moreover, NEK2 knockdown by shRNA in HCT116 cells significantly downregulated the level of active RhoA and Rac1, while active Cdc42 levels remained unchanged (Figure 4C). These data indicated that NEK2 activates RhoA and Rac1 but not Cdc42.

To investigate whether NEK2-mediated Rho GTPases activation was associated with the interaction between NEK2 and RhoGDI1, NEK2-overexpressing DLD1 cells were stably transfected with the mCherry only or the mCherry-RhoGDI1 aa 112–134 fragment. As expected, RhoGDI1 aa 112–134 suppressed the activation of RhoA and Rac1 by NEK2 overexpression (Figure 4D), suggesting that the interaction of NEK2 with RhoGDI1 was indispensable for the activation of RhoA and Rac1 induced by NEK2.

### 3.4. Phosphorylation of RhoGDI1 by NEK2 Enhances Cancer Cell Proliferation, Migration and Invasion

Given that the activity of Rho GTPases is crucial for cancer cell proliferation and motility [2,38,39], and NEK2 enhanced the activity of RhoA and Rac1, we investigated the impact of NEK2 on the proliferation and metastatic behavior of colon cancer cells. WST-8 assay showed that treatment of HCT116 cells with the NEK2 inhibitor resulted in a significant reduction in cell proliferation compared to cells treated with DMSO (Figure 5A). Moreover, the wound-healing assay and invasion assay show that the NEK2 inhibitor suppressed the migration (Figure 5B) and invasion (Figure 5C) of HCT116 cells. These results suggest that the activity of NEK2 is necessary for the proliferation, migration and invasion of cancer cells. To investigate the role of NEK2-mediated phosphorylation of RhoGDI1 at Ser174 in regulating cell migration and invasion, DLD-1 cells were transfected with NEK2 along with either the RhoGDI1 WT or S174A substitution mutant. Cell migration and invasion were enhanced by NEK2 overexpression when co-transfected with RhoGDI1 WT, whereas these effects were abrogated in cells expressing the RhoGDI1 S174A mutant (Figure 5D,E). These results indicate that the phosphorylation of RhoGDI1 at a Ser174 residue is essential for promoting the proliferation, migration and invasion of cancer cells.

To verify whether NEK2-mediated cancer cell malignant behaviors are involved in its interaction with RhoGDI1, we stably expressed HA-NEK2 along with the mCherry-RhoGDI1 aa 112–134 fragment in DLD-1 cells. Subsequently, we conducted WST-8, wound-healing and invasion assays. The results demonstrate that the overexpression of NEK2 slightly increased cell growth (Figure 6A) and significantly enhanced the migration (Figure 6B) and invasion (Figure 6C) of DLD-1 cells. However, these effects were reversed upon overexpression of the mCherry-RhoGDI1 aa 112–134 fragment (Figure 6). Also, we transfected GFP-RhoGDI1 aa 112–134 in HCT116 cells stably expressing control shRNA or two NEK2 shRNAs and performed a proliferation assay, wound-healing assay and invasion assay in the cells (Appendix A). Consistent with results from the overexpression system, transfection of GFP-RhoGDI1 aa 112–134 fragment reduced cell proliferation and mobility in the presence of NEK2, suggesting that the interaction between NEK2 and RhoGDI1 is critical for enhancing cancer cell growth, migration and invasion.

### 3.5. Inhibition of Interaction Between NEK2 and RhoGDI1 Attenuates Cancer Growth and Metastasis

To assess the effect on the interaction of NEK2 with RhoGDI1 for cancer cell growth in vivo, DLD-1 cells which overexpress the mCherry only or mCherry-RhoGDI1 aa 112–134 fragment with or without NEK2 were subcutaneously injected into the mice. The tumor weight and volume from NEK2-overexpressing cells significantly increased compared to those from the control cells. However, this NEK2-mediated tumor growth enhancement was effectively reversed by the expression of the RhoGDI1 aa 112–134 fragment (Figure 7A–C). Furthermore, the expressions of Ki-67 (a proliferation marker) and CD31 (an endothelial cell marker) in the tumors from the NEK2-overexpressing cells were enhanced compared with the control cells. Notably, the expression of the RhoGDI1 aa 112–134 fragment reversed these increases in Ki-67 and CD31 expression induced by NEK2 (Figure 7D). These findings indicate that the interaction of NEK2 with RhoGDI1 plays a critical role in promoting tumor angiogenesis and enhancing tumor growth in vivo.

Next, we investigated the effects on the interaction of NEK2 with RhoGDI1 for distant metastasis in vivo. Mice injected with NEK2-expressing cells exhibited numerous metastatic lung nodules compared to those injected with mock vector-expressing cells. Remarkably, the NEK2-mediated lung metastasis was significantly attenuated by the expression of the RhoGDI1 aa 112–134 fragment (Figure 7E,F). These findings indicate that the interaction of NEK2 with RhoGDI1 is essential for NEK2-induced metastasis.

## 4. Discussion

NEK2 is overexpressed in a variety of human cancers, such as colon cancer [28], breast cancer [29], gastric cancer [30], hepatocellular carcinoma [40] and pancreatic cancer [41]. Its high expression is related to enhanced tumor progression and poor prognosis of patients with cancer [28,29,30,40,41]. NEK2 is a serine/threonine kinase and plays multiple roles during mitosis [18,42,43,44]. For example, NEK2 phosphorylates centrosome linker proteins, cNAP1 (centrosomal NEK2-associated protein 1) and rootletin, which induces centrosome duplication and separation [42,43]. NEK2 localizes at the kinetochore where it phosphorylates Hec1 at Ser165. This phosphorylation enables Hec1-pS165 to recruit and interact with the MAD1 protein, thereby facilitating the activation of the spindle assembly checkpoint (SAC) signaling in response to misaligned chromosomes [18,44]. Recent studies have highlighted NEK2’s role in regulating the migration and invasion of cancer cells. [23,25,27]. Downregulation or inhibition of NEK2 suppresses the migration and invasion of breast cancer cells [23]. NEK2 increases hepatocellular carcinoma (HCC) cell invasion by the epithelial–mesenchymal transition (EMT), and the various signaling pathways, including Wnt, NF-κB and focal adhesion which are predicted as downstream of NEK2 by gene expression microarray analysis [25]. MiR-1299 transcriptionally repressed NEK2 in prostate cancer cells, attenuating the proliferation, invasion and migration [27]. Consistent with these, NEK2 overexpression and knockdown experiments demonstrated that NEK2 promoted the proliferation, migration and invasion of colon cancer cells (Figure 5 and Figure 6 and Appendix A).

Although recent findings have demonstrated the involvement of NEK2 in metastatic activities of cancer cells, its precise underlying remains poorly defined. In this study, we provide evidence suggesting a novel mechanism by which NEK2 regulates cancer cell growth and motility. We identified NEK2 as a binding partner of RhoGDI1, and found that the RhoGDI1 aa 112–134 region was essential for the interaction with NEK2 (Figure 2). Disruption of the NEK2–RhoGDI1 interaction by overexpressing RhoGDI1 aa 112–134 reversed the enhanced proliferation, migration, and invasion induced by NEK2 (Figure 6). Furthermore, in vivo experiments demonstrated that NEK2 overexpression enhanced the tumor growth, angiogenesis and lung metastasis of DLD-1 cells, whereas the overexpression of RhoGDI1 aa 112–134 restored the enhancement promoted by NEK2 (Figure 7). Therefore, enhanced malignant phenotypes promoted by NEK2 may be attributed to its interaction with RhoGDI1.

NEK2 is relatively overexpressed in HCT116 cells compared to DLD-1 cells (Figure 1C). Overexpression of NEK2 in DLD-1 cells increased the basal activities of RhoA and Rac1, while NEK2 knockdown in HCT116 cells decreased the basal activities of RhoA and Rac1 (Figure 4A,C). Interestingly, upon treatment of MDA-MB-231 and MCF7 cells with insulin-like growth factor-1 (IGF), the increased cell viability and phosphorylation of ERK were reversed by NEK2 knockdown [45]. Therefore, it would be meaningful for future studies to investigate the relationship between NEK2-mediated phosphorylation of RhoGDI1 and changes in cell phenotypes induced by stimulation with growth factors such as IGF.

We verified the effects on cells of the RhoGDI1 aa 112–134 fragment interrupting the interaction between NEK2 and RhoGDI1 (Figure 6, Appendix A). When we conducted experiments with HCT116 cells stably expressing two NEK2 shRNAs, the cells exhibited very little mobility and invasiveness (Appendix A). If these cell lines were used for in vivo experiments, it would be difficult to observe metastasis and determine the effect of the RhoGDI1 aa 112–134 fragment on the metastatic activities of the cells. In contrast, when using an overexpression system, mock vector-transfected DLD-1 cells exhibited mobility and invasiveness, and NEK2 overexpressing DLD-1 cells showed increased migration and invasion compared to control DLD-1 cells. Therefore, to test whether the interaction between NEK2 and RhoGDI1 is related to in vivo tumor growth and metastasis, we used NEK2-overexpressing cell lines. Our animal studies demonstrated that the increase in tumor growth and lung metastasis due to NEK2 overexpression was reduced by the expression of the RhoGDI1 fragment.

Phosphorylation of RhoGDI1 typically reduces its affinity for Rho GTPases, resulting in increased dissociation of Rho GTPases, and their subsequent activation [46]. For instance, phosphorylation of RhoGDI1 by PKCα, Src or PAK1 reduced its association with specific Rho GTPase [33,34,35,47]. PKCα is known to phosphorylate Ser34 and Ser96 of RhoGDI1. Phosphorylation of RhoGDI1 on Ser34 specifically decreased the affinity to RhoA but not Rac1 [33]. In a subsequent study, however, cholecystokinin (CCK) stimulated the activation of PKCα in pancreatic acini, which mediated the phosphorylation of RhoGDI1 at Ser96, releasing both cytosolic RhoA and Rac1 associated with RhoGDI1, and facilitating Rho GTPases signaling [34]. Additionally, Src decreased the complex of RhoGDI1 with RhoA, Rac1 or Cdc42 by phosphorylating Tyr27 and Tyr156 on RhoGDI1 [47]. PAK1 phosphorylated RhoGDI1 at Ser101 and Ser174, promoting the dissociation and subsequent activation of Rac1, but not RhoA [35]. Thus, it seems that the affinity of RhoGDI1 and Rho GTPase can be selectively controlled as various kinases phosphorylate specific residues.

The present study demonstrated that NEK2 directly phosphorylated RhoGDI1 at Ser174 residue in vitro and in cells (Figure 3). Upon EGF stimulation, RhoGDI1 phosphorylated at Ser174 binds to 14-3-3, promoting the dissociation of RhoA, Rac1 and Cdc42, which leads to their activation and results in cancer cell invasion and breast cancer metastasis [10]. Since NEK2 overexpression significantly increased the interaction of RhoGDI1 with 14-3-3 (Figure 3G), the phosphorylation of RhoGDI1 at Ser174 by NEK2 may enhance the dissociation of RhoGDI1 from RhoA, Rac1 and Cdc42. Interestingly, our results show that the activities of RhoA and Rac1 were increased by NEK2 overexpression, while their activities were reduced by the inhibition of NEK2 by shRNA or a pharmacological inhibitor (Figure 4). Therefore, the phosphorylation of RhoGDI1 at Ser174 by NEK2 is not sufficient for the activation of RhoGTPases, but it clearly contributes to the activation of specific RhoGTPases. NEK2-mediated activation of RhoGTPases may also require the involvement of other factors, such as specific RhoGEFs. Notably, the overexpression of the RhoGDI1 S174A mutant effectively suppressed NEK2-mediated migration and invasion of DLD-1 cells (Figure 5D,E), suggesting that NEK2 enhances cell migration and invasion, at least in part, through its phosphorylation of RhoGDI1 at Ser174.

Meanwhile, Rho GTPases are also known to function in mitosis [48,49,50,51,52,53]. At mitosis entry, Rho-associated kinase (ROCK) activation following increased RhoA activity results in cortical retraction for mitotic cell rounding [48]. During early mitosis, active Rho regulates spindle assembly by interacting with mDia1 in Rat-2 cells, and Cdc42-mDia3 signaling is required for attachment of microtubules to kinetochores in HeLa cells [49,50]. Later in mitosis, RhoA is involved in cytokinesis, where it spatiotemporally regulates contractile ring assembly, resulting in the formation of the cleavage furrow [51,52,53]. Furthermore, NEK2 may be a mitotic regulator that is involved in diverse cell cycle events [42,43,44]. Thus, further studies are necessary to understand the molecular mechanism underlying NEK2-mediated RhoGTPases activation during mitosis.

## 5. Conclusions

In conclusion, the current study elucidated a novel molecular mechanism by which NEK2 regulates the metastatic behaviors of colon cancer cells by activating Rho GTPases through phosphorylating RhoGDI1 (Figure 8). Furthermore, these results pose the possibility that targeting the interaction of RhoGDI1 and NEK2 could be a promising therapeutic strategy for cancer treatment.

## Figures and Tables

**Figure 1 cells-13-02072-f001:**
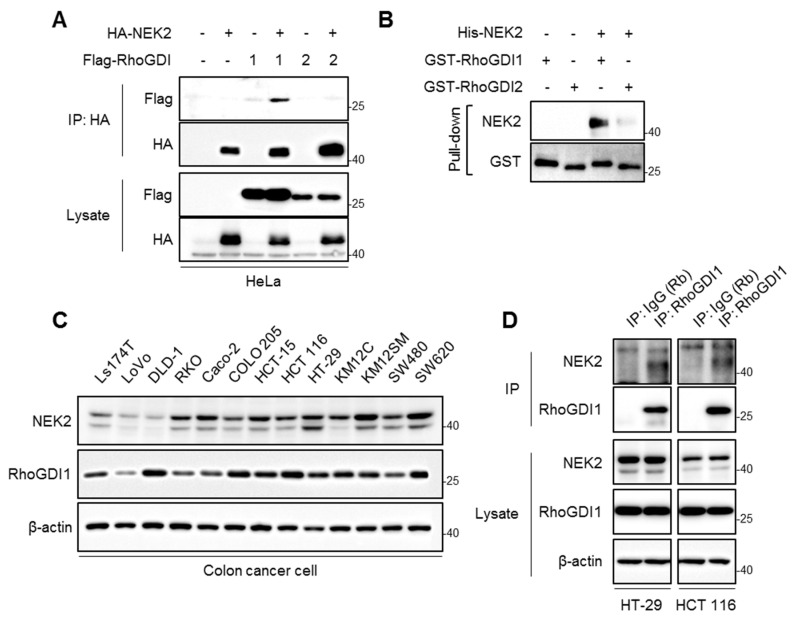
NEK2 interacts with RhoGDI1 but not RhoGDI2. (**A**) HeLa cells were co-transfected with HA-tagged NEK2 and Flag-tagged RhoGDI1 or RhoGDI2. Cell lysates were subjected to immunoprecipitation using HA antibody, followed by Western blotting with HA and Flag antibodies. (**B**) GST pull-down assay was conducted using recombinant His-tagged NEK2 and GST-tagged RhoGDI1 or RhoGDI2. (**C**) Western blot analysis was performed to assess NEK2 and RhoGDI1 expression in human colon cancer cell lines. (**D**) Cell lysates from HT-29 and HCT116 were immunoprecipitated with either IgG or RhoGDI1 antibodies, followed by Western blot analysis using indicated antibodies.

**Figure 2 cells-13-02072-f002:**
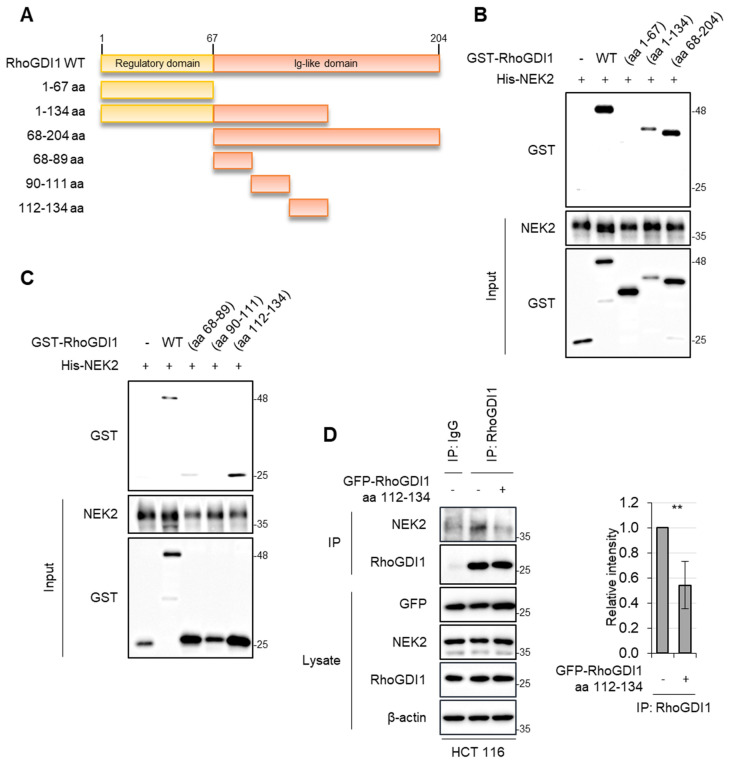
The requirement of the RhoGDI1 aa 112–134 region for its interaction with NEK2. (**A**) Schematic diagram of RhoGDI1 WT and 6 truncated fragments. (**B**,**C**) Purified GST-tagged RhoGDI1 WT or truncation fragments along with recombinant His-NEK2 were subjected to His pull-down assay. His pull-down samples were analyzed by WB using NEK2 and GST antibodies. (**D**) HCT116 cells were transfected with the mock vector or GFP-tagged RhoGDI1 aa 112–134 fragment. Cell lysates were immunoprecipitated with either IgG or RhoGDI1 antibodies, followed by Western blot analysis using indicated antibodies (left). Relative band intensities (NEK2/RhoGDI1) were quantified using Image J and shown as a graph (right). Quantitative data represent the mean ± S.D. (*n* = 3). ** *p* < 0.01.

**Figure 3 cells-13-02072-f003:**
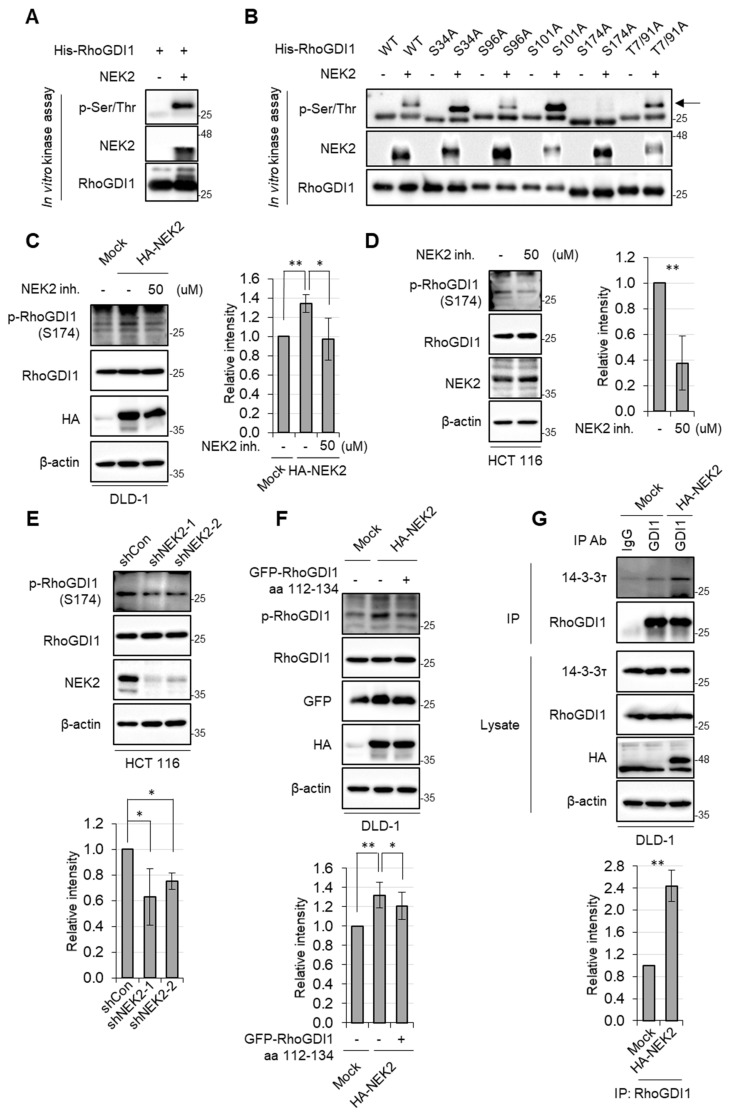
NEK2 phosphorylates RhoGDI1 at Ser174. (**A**) Purified His-RhoGDI1 were subjected to in vitro kinase assay with recombinant active NEK2 (+) or without NEK2 (−). Samples were then analyzed by WB using indicated antibodies. (**B**) Purified His-RhoGDI1 WT and substituted mutants (S34A, S96A, S101A, S174A and T7/91A) were subjected to in vitro kinase assay with recombinant active NEK2 (+) or without NEK2 (−), followed by WB analysis using indicated antibodies. The arrow indicates phosphorylated RhoGDI1. (**C**) DLD-1 cells were stably transfected with HA-NEK2. (**C**,**D**) DLD-1 and HCT116 cells were incubated with 50 nM of NCL 00017509 (NEK2 inhibitor) in serum-free media for 24 h, followed by WB analysis using indicated antibodies (left). Relative band intensities (p-RhoGDI1/RhoGDI1) were quantified using Image J and shown as a graph (right). (**E**) HCT116 cells stably expressing control shRNA or two NEK2 shRNAs were incubated in serum-free media for 24 h. Cell lysates were subjected to WB analysis using indicated antibodies (upper). Relative band intensities (p-RhoGDI1/RhoGDI1) (lower). (**F**) DLD-1 cells stably expressing mock or HA-NEK2 were transiently transfected with mock or GFP-RhoGDI1 aa 112–134 expressing vector. Cells were incubated in serum-free media for 24 h. Cell lysates were analyzed by WB using indicated antibodies (upper). Relative band intensities (p-RhoGDI1/RhoGDI1) (lower). (**G**) DLD-1 cells stably expressing mock or HA-NEK2 were incubated in serum-free media for 24 h. IP analysis was performed with DLD-1 cell lysates and a RhoGDI1 antibody, followed by WB analysis using indicated antibodies (upper). Relative band intensities (14-3-3 tau/RhoGDI1) (lower). Quantitative data represent the mean ± S.D. (*n* = 3). * *p* < 0.05; ** *p* < 0.01.

**Figure 4 cells-13-02072-f004:**
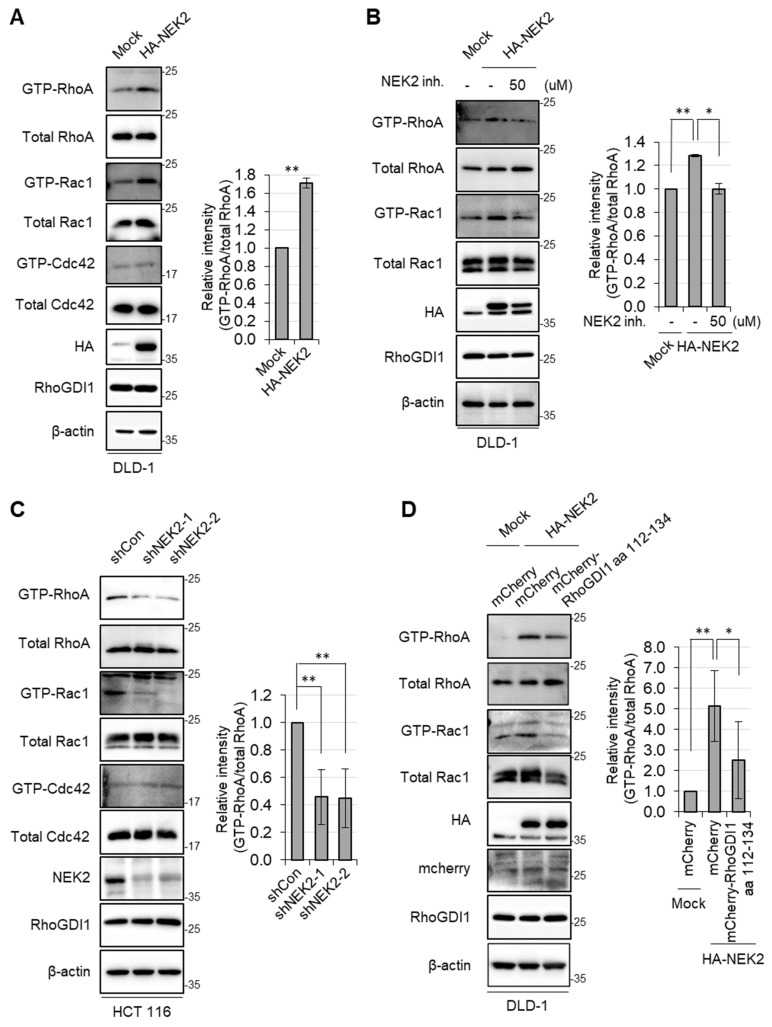
NEK2 facilitates the activation of RhoA and Rac1 by interaction with RhoGDI1. (**A**) DLD-1 cells stably expressing mock or HA-NEK2 were incubated in serum-free media for 24 h. A pull-down assay was performed to assess the levels of active RhoA and Rac1/Cdc42, as described in the Materials and Methods Section (left). (**B**) DLD-1 cells stably expressing mock or HA-NEK2 were treated with 50 nM of NCL 00017509 in serum-free media for 24 h, followed by pull-down assay and WB analysis (left). (**C**) HCT116 cells stably expressing control shRNA or two NEK2 shRNAs were incubated in serum-free media for 24 h and subjected to pull-down assay and WB analysis (left). (**D**) HA-NEK2 expressing DLD-1 cells were stably transfected with mock or mCherry-RhoGDI1 aa 112–134 expressing vector. Cells were incubated in serum-free media for 24 h, followed by pull-down assay and WB analysis (left). Relative band intensities (GTP-RhoA/total RhoA) were measured by Image J and shown as a graph (right). Quantitative data represent the mean ± S.D. (*n* = 3). * *p* < 0.05; ** *p* < 0.01.

**Figure 5 cells-13-02072-f005:**
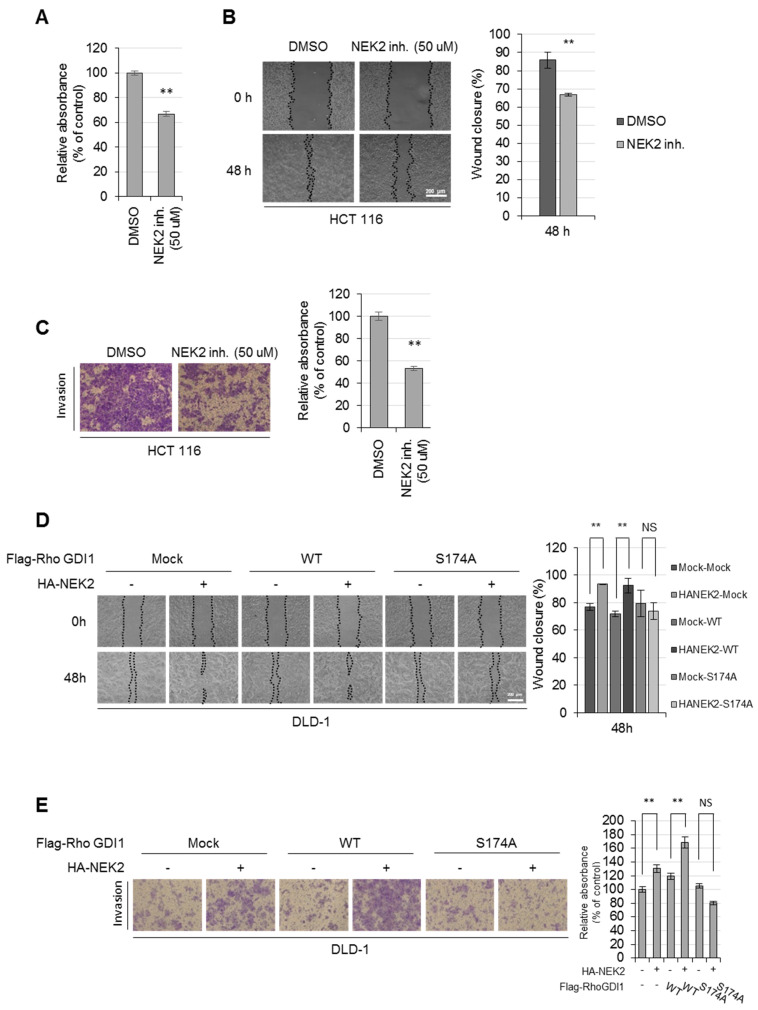
NEK2 promotes migration and invasion of colon cancer cells by phosphorylating RhoGDI1. (**A**–**C**) HCT116 cells were incubated with 50 nM of NCL 00017509, NEK2 inhibitor. (**A**) The viability of treated cells was assessed using a WST-8 assay. The graph represents the relative percentages of proliferating cells compared to untreated control. (**B**) Cell migration was evaluated using wound-healing assay at indicated time point. Representative images of migrating cells obtained at 48 h after wound formation (left). Scale bar = 200 μm. The migration was quantified by calculating the cell-covered area using Image J (right). (**C**) Cells were incubated in serum-free media for 24 h and subjected to transwell invasion assay. Representative images (100×) of invading cells are shown on the left, and the relative percentages of invasion are presented on the right. (**D**,**E**) DLD-1 cells stably expressing mock or HA-NEK2 were transiently transfected with Flag-RhoGDI1 WT or Flag-RhoGDI1 S174A. (**D**) Cell migration was evaluated using wound-healing assay at indicated time point. Representative images of migrating cells are shown on the left, and the percentage of wound closure is depicted on the right. Scale bar = 200 μm. (**E**) Cells were incubated in serum-free media for 24 h and subjected to transwell invasion assay. Migrating or invading cells were shown in representative images (left) or the relative percentages of invasion (right). Quantitative data represent the mean ± S.D. (*n* = 3). ** *p* < 0.01; NS, non-significant.

**Figure 6 cells-13-02072-f006:**
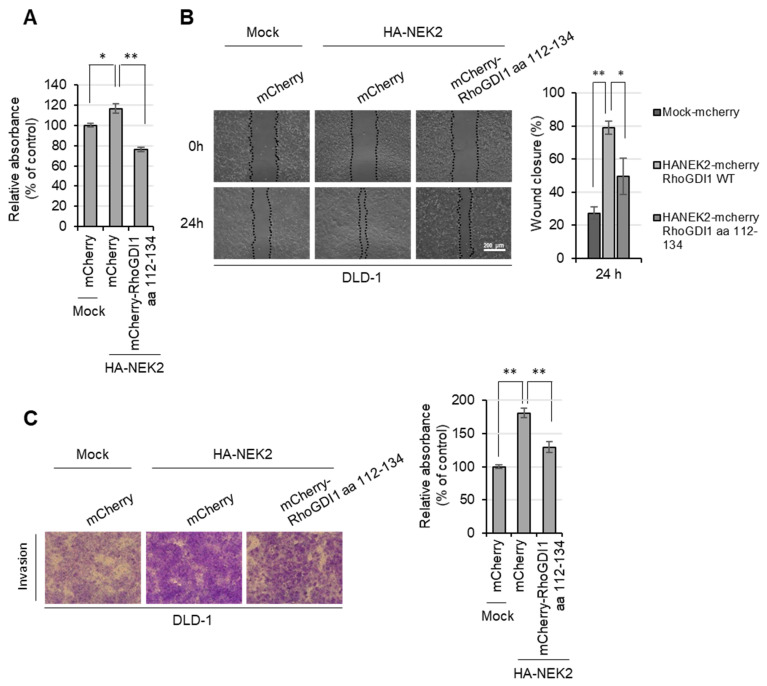
Interaction of NEK2 with aa 112–134 on RhoGDI1 is crucial for promoting proliferation, migration and invasion of colon cancer cells. DLD-1 cells expressing mock or HA-NEK2 were stably transfected with mCherry or mCherry-RhoGDI1 aa 112–134. (**A**) Cells were subjected to WST-8 assay. The graph represents the relative percentages of proliferating cells. (**B**) Migration of indicated cells was evaluated by wound-healing assay at each time point. Representative images of migrating cells obtained at 24 h after wound formation (left). Scale bar = 200 μm. The migration was quantified by calculating the cell-covered area using Image J (right). (**C**) Cells were incubated in serum-free media for 24 h and then subjected to transwell invasion assay. Representative images (100×) of invading cells are displayed on the left, and the relative percentages of invasion are quantified on the right. Quantitative data represent the mean ± S.D. (*n* = 3). * *p* < 0.05; ** *p* < 0.01.

**Figure 7 cells-13-02072-f007:**
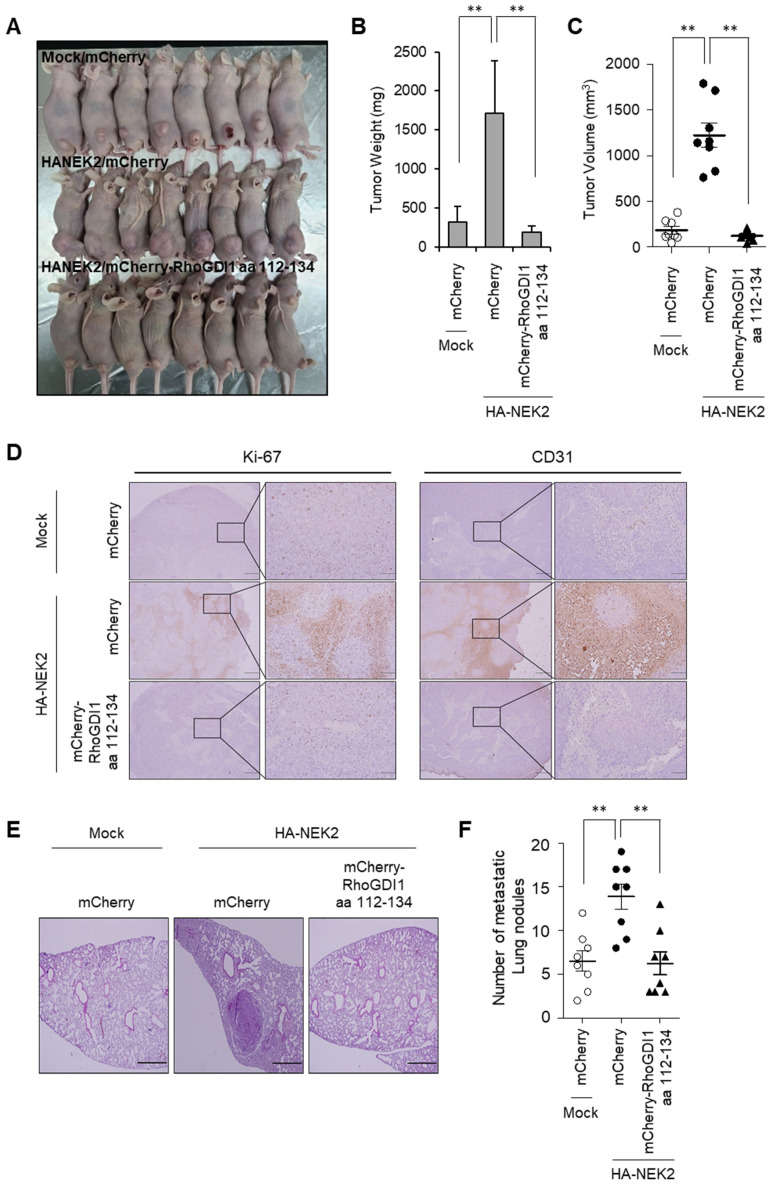
NEK2 promotes tumor growth and metastasis of colon cancer through its association with RhoGDI1. (**A**–**D**) DLD-1 cells expressing either mCherry or mCherry-RhoGDI1 aa 112–134 with or without HA-NEK2 were subcutaneously inoculated into the mice (5 × 10^6^/mouse). (**A**) Representative image of the tumors from each group of mice. (**B**) Measurement of tumor weights. (**C**) Measurement of tumor volumes, following procedures described in the Materials and Methods Section. (**D**) Tumor sections of each mouse were stained with anti-Ki67 or anti-CD31 antibody to assess proliferation and angiogenesis, respectively. Scale bar = 500 μm. Insets display accumulation of Ki-67 or CD31. Scale bar = 100 µm. (**E**,**F**) Indicated DLD-1 were intravenously inoculated into the mice (2 × 10^6^/mouse). (**E**) Representative images of H&E-stained lung tissues of the mice injecting DLD-1 cell lines. Scale bar = 500 μm. (**F**) The number of metastatic nodules was counted per lung tissue and represented in the scatter plots. Quantitative data represent the mean ± S.D. (*n* = 8). ** *p* < 0.01.

**Figure 8 cells-13-02072-f008:**
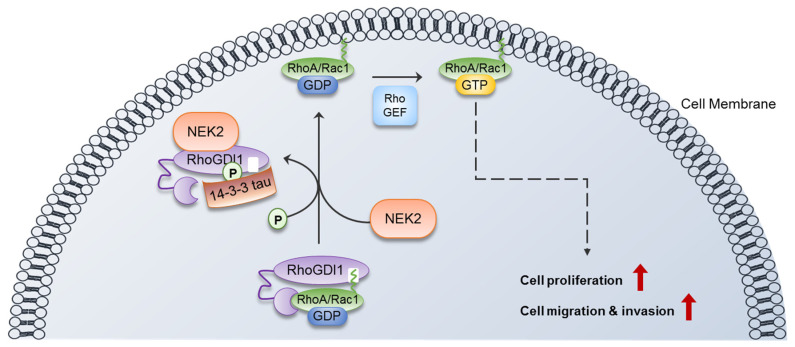
The schematic diagram shows that NEK2 induces proliferation and metastatic behaviors of cells. The red arrows indicate increase.

## Data Availability

The data supporting the findings of this study are available from the corresponding author upon reasonable request.

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
