# Peer review of "NEK2 Phosphorylates RhoGDI1 to Promote Cell Proliferation, Migration and Invasion Through the Activation of RhoA and Rac1 in Colon Cancer Cells"

_cells, 2024, doi:10.3390/cells13242072_

Round 1
Reviewer 1 Report
Comments and Suggestions for Authors
This work by Lim et al demonstrates a direct interaction between NEK2 and RhoGDI1. They show that RhoGDI1 is phosphorylated by NEK2, leading to increased activation of Rac1 and RhoA and increased migratory/invasive potential of the cells. In general, this is an appropriately designed study with convincing results. However, I have a some concerns which can be easily addressed.
1. My first concern is in reference to the presentation of data. Some blots, where the observed result is not binary, should be quantified and a graph of independent repeats with statistical analysis presented alongside the blot. These include figures 2D, 3C, 3D, 3E, 3F, 3G, 4A, 4B, 4C, and 4D.
2. Secondly, I have concerns about the use of the RhoGDI aa 112-134 mutant. How can the authors be sure that the negative effects of this mutant are dependent on interrupting the interaction between NEK2 and RhoGDI? It is possible that this mutant has a dominant negative effect and would still impair proliferation, migration and invasion in the absence of NEK2 due to the disruption of other proteins interacting with RhoGDI. To demonstrate whether this is the case, the authors should perform an experiment expressing this mutant in the absence of NEK2 (NEK2 null cells or NEK2 knockdown cells) to demonstrate that the negative effects of its expression only occurs in the presence of NEK2. If it has a general dominant negative effect then this should be shown and acknowledged as a potential explanation for the inhibitory effect on tumor growth in figure 7a.
3. Why did the authors use an overexpression system in the in vivo experiment rather than the knockdown of endogenous NEK2 in a cell line with high NEK2 expression? The authors should explain in the manuscript why they chose this approach when a knockdown study would be more physiologically relevant.
Minor concerns
A) The title should state that this role for NEK2 and RhoGDI is in colon cancer as the authors haven't demonstrated a cancer wide effect.
B) In line 44 the authors state that GEFs activate Rho GTPases by converting GDP to GTP. This is not correct. GEFs facilitate the activation of GTPases by stabilizing the nucleotide free conformation of the GTPase resulting in the dissociation of GDP. GTP then freely associates to the nucleotide pocket.
C) In line 186 the authors state that NEK2 "exhibited high" expression in colon cancer cell lines. How are the authors defining "high"?
D) In line 191, "in vivo" should be changed to "in cells".
Comments on the Quality of English LanguageIn general, the English writing is good and easy to follow. However, there are some places where statements are ambiguous or the word order confused, which changes the meaning of the sentence. This is most prominent in the introduction and discussion. For example, in line 46 the sentence should read "intrinsic hydrolysis of GTP".
Author Response
We are submitting a revised manuscript (cells-3204362) entitled “NEK2 promotes cancer cell proliferation, migration and invasion through the interaction with RhoGDI1.” for publication in Cells. We greatly appreciate the suggestions and comments of the reviewers. We have addressed their concerns and modified the manuscript accordingly. The modified manuscript was highlighted in red for ease of identification by the reviewers.
The point-by-point responses to the reviewer's comments are described in the attached file.

Reviewer 2 Report
Comments and Suggestions for Authors
This is a nice contribution by Lim and co-workers. The authors convincingly show that RhoGDI1 is a physiological substrate of Nek2. Mapping studies showed the importance of region 112-134 of RhoGDI for this interaction. The manuscript further studies the molecular and key molecular players involved. The experimentation is well justified and conducted and interpreted in a correct way. In summary, the data suggest that Nek2 binding and phosphorylation of RhoGDI result in the activation of RhoA and RAc1 and a high cellular proliferation, cell migration and cell invasion. All together the data are convincing and the level of novelty is good. I can only recommend publication.
Author Response
Thank you for your positive feedback on our work. We appreciate your recognition of RhoGDI1 as a physiological substrate of Nek2 and the importance of the RhoGDI region 112-134. We are pleased that you found our experiments well justified and correctly interpreted. Your recommendation for publication is greatly appreciated.
Reviewer 3 Report
Comments and Suggestions for Authors
The manuscript “NEK2 phosphorylates RhoGDI1 to promote cancer cell proliferation, migration and invasion through the RhoA and Rac1 activation” by Jeewon Lim, Yo Sep Hwang, Jong-Tae Kim, Hyang Ran Yoon, Hyo-Min Park, Jahyeong Han, Taeho Kwon, Kyung Ho Lee, Hee Jun Cho and Hee Gu Lee studies the relations between RhoGDI1 and NEK2, using biochemical tools and in vivo assays. This study is well designed and the results present interest to searchers in the field of GTPases and of cancer.
However, I have several concerns that I will expose hereafter.
Major concerns:
- Figure 2B is a big mess. In fact, it is not a GST pull down as described. It seems to be a Ni-column His-tag pulldown with NEK2 as a bait to catch GST-constructs with different fragments of RhoGDI1. It has to be totally rewritten, changing the legend, the figure itself (input must include NEK2), the text (L 201-213) and also include the method in its chapter (no mention of Nickel column).
- Figure 3C: The enhancement of phosphorylation of RhoGDI1 is not very obvious. The blots are of poor quality, and I suggest to the authors to make a densitometric analysis of their blots and propose a histogram with error bars (the same would be interesting for fig4 because the densitometric values have no statistical evaluation, and we don’t know the number of experiments). Also, in the text (L237), “notably” is too affirmative.
- In the same way, in figure 4A and B, the value corresponding to RhoA activation state are very discrepant between the two figures, although the experimental conditions are identical. Here, a statistical value is indispensable (taking into account that this is not the case for Rac1).
- Interestingly, the use of shRNA against NEK2 or NEK2 inhibitor induce a decrease in the basal state of activation of RhoA and Rac1. This is not explained in the discussion and should be developed by the authors. The question is that NEK2 could have a role in the homeostasis of RhoA and Rac1 activity. Quid of the regulation of the activation of Rac1 and RhoA with cytokines or growth factors? It would be interesting to develop this part by adding experiments with stimulation of the cells as in the Boyden assays.
- In figure 5D and E, it would be necessary (to continue with the preceding remark) to add a Boyden experiment using only S174A mutant without NEK2 overexpression, to see if there is an inhibition of basal ability of cells to migrate or invade in these conditions.
- Finally, is there any difference in the affinity for RhoA, Rac1 and Cdc42 of this S174A mutant. Could it be done? It would give a very strong clue to understand the mechanism underlying these cross-regulations.
Minor concerns:
- L57-60: The reference used concerns histone phosphorylation and has no relation with RhoGDI. I suggest changing this reference (10).
- L208-210: there is a mistake in the text concerning the use of fragments of RhoGDI1. Authors mention 112-134 two times where it should be the 90-111 the second occurrence. Also, the 68-89 fragment shows a mild affinity to NEK2, and this is not mentioned.
- The use of the term mutants for fragments of RhoGDI1 is not correct. These are not mutants, but fragments. Please modify.
- L409, “We” should be in upper case letter.
- L441-444: The sentence has no meaning. Stimulation is repeated two times, please modify.
- Finally, the last scheme is interesting but lacks the presence of 14-3-3tau. Because the authors discussed its implication, I suggest adding it in the representation.
Comments on the Quality of English LanguageThere are few sentences to be modified. These were pointed out in the comments
Author Response
We are submitting a revised manuscript (cells-3204362) entitled “NEK2 promotes cancer cell proliferation, migration and invasion through the interaction with RhoGDI1.” for publication in Cells. We greatly appreciate the suggestions and comments of the reviewers. We have addressed their concerns and modified the manuscript accordingly. The modified manuscript was highlighted in red for ease of identification by the reviewers.
The point-by-point answers to each reviewer’s comments are described in the attached file.

Round 2
Reviewer 1 Report
Comments and Suggestions for Authors
All my concerns have been addressed.
Author Response
Thank you for your positive feedback on our work. Your recommendation for publication is greatly appreciated.
Reviewer 3 Report
Comments and Suggestions for Authors
I thank the authors for answering to my requests. I think the readability is now enhanced. Globally, they answered to each questions, but I still have two concerns.
1- In line 435-6, the sentence should be "Overexpression of NEK2 in DLD-1 cells increased the BASAL activities of RhoA and Rac1, while NEK2 knockdown in HCT116 decreased the BASAL activities of RhoA and Rac1 (Figure 4 A,C)" in order to strengthen the message that there is no stimulation here.
2- Concerning my comment#5, I am not satisfied by the answer provided by the authors. They agree that a mutant of Ser174 of RhoGDI1 is likely to have an impact. We would evaluate this impact by adding the experiment with the mutant alone (even if the impact is so important that NEK2 would not add any modification in the motility). This information is very important and constitute an important control, because without it, one can't evaluate the influence of NEK2 in these phenotypes. Please add the experiment.
Author Response
I thank the authors for answering to my requests. I think the readability is now enhanced. Globally, they answered to each questions, but I still have two concerns.
1- In line 435-6, the sentence should be "Overexpression of NEK2 in DLD-1 cells increased the BASAL activities of RhoA and Rac1, while NEK2 knockdown in HCT116 decreased the BASAL activities of RhoA and Rac1 (Figure 4 A,C)" in order to strengthen the message that there is no stimulation here.
Response: We agree with the reviewer’s comment. The statement has been revised to “Overexpression of NEK2 in DLD-1 cells increased the basal activities of RhoA and Rac1, while NEK2 knockdown in HCT116 decreased the basal activities of RhoA and Rac1 (Figure 4A,C).” (Line 435-6).
2- Concerning my comment#5, I am not satisfied by the answer provided by the authors. They agree that a mutant of Ser174 of RhoGDI1 is likely to have an impact. We would evaluate this impact by adding the experiment with the mutant alone (even if the impact is so important that NEK2 would not add any modification in the motility). This information is very important and constitute an important control, because without it, one can't evaluate the influence of NEK2 in these phenotypes. Please add the experiment.
Response: As the reviewer’s insightful comments, we have now added the experiments including S174A mutant alone to assess the motility effect of NEK2 in Figure 5D,E. (Line323-325)

Round 3
Reviewer 3 Report
Comments and Suggestions for Authors
Thank you to the authors for providing the demanded experiments. It is now more clear and easy to understand.
Good luck for future research.